# Japanese Cross-Sectional Multicenter Survey (JAMS) of Oral Appliance Therapy in the Management of Obstructive Sleep Apnea

**DOI:** 10.3390/ijerph16183288

**Published:** 2019-09-06

**Authors:** Kentaro Okuno, Akifumi Furuhashi, Shuhei Nakamura, Hiroshi Suzuki, Takehiro Arisaka, Hitoshi Taga, Masataka Tamura, Haruto Katahira, Minoru Furuhata, Chisato Iida

**Affiliations:** 1Department of Geriatric Dentistry, Osaka Dental University, Osaka 540-0008, Japan; 2Division for Oral and Facial Disorders, Osaka University Dental Hospital, Suita 565-0871, Japan; 3Department of Oral and Maxillofacial Surgery, Aichi Medical University, Nagakute 480-1195, Japan; 4Dental Clinic for Sleep Disorders (Apnea and Snoring), Oral and Maxillofacial Rehabilitation, University Hospital of Dentistry, Tokyo Medical and Dental University, Tokyo 113-8510, Japan; 5Division of Oral Function and Rehabilitation, Department of Oral Health Science, Nihon University School of Dentistry at Matsudo, Matsudo 271-0061, Japan; 6Ota Memorial Sleep Center, Sleep Surgery Center, Kawasaki 210-0024, Japan; 7Dentistry and Oral Surgery, JR Tokyo General Hospital, Tokyo 151-8528, Japan; 8Department of Dentistry and Maxillo-Facial Surgery, Komatsu Hospital, Neyagawa 572-0015, Japan; 9Katahira Dental Clinic, Tokyo 151-0053, Japan; 10Furuhata Sleep-disordered Breathing Research Institute, Furuhata Dental Clinic, Tokyo 107-0052, Japan; 11Snore & Sleep Apnea Treatment Center, Nippon Dental University Hospital Internal Medicine, Tokyo 173-8610, Japan; 12Iida Dental Clinic, Soka 340-0013, Japan

**Keywords:** obstructive sleep apnea (OSA), oral appliance (OA), multicenter survey

## Abstract

We conducted a multicenter survey for oral appliance (OA) therapy to grasp and analyze the current situation of OA therapy, including efficacy, side effects, and follow-up, in Japan. The Japanese cross-sectional multicenter survey (JAMS) for obstructive sleep apnea (OSA) was undertaken by patients in 10 institutions associated with oral appliance therapy during two years, from 2014 to 2015, retrospectively. Age, sex, body mass index (BMI), baseline apnea–hypopnea index (AHI), OA type, adjustment, adverse reactions with OA, and AHI with OA were elicited from the patient clinical record. The number of included OSA patients was 3217. The number of patients with OA therapy was 2947. We evaluated 1600 patients for the OA treatment. The patients treated with OA, both male and female, were most commonly in their 50s. In terms of OSA severity, snoring was 2.3%, mild was 38.5%, moderate was 39.9%, and severe was 19.3%. The use of mono bloc appliance was 91.8%, bi bloc appliance was 7.9%, and tongue-retaining device (TRD) was 0.3%. After OA treatment, AHI decreased from 22.4/h to 9.3/h. The AHI reduction rate with OA was 52.0%. The rate of AHI <5 with OA was 35.6%. Adverse reactions developed in 14.7% of the subjects. OA reassessment was conducted for 54.3%. This study revealed the current situation of efficacy and side effects of OA therapy, and the problem that the reassessment rate of OA was low in Japan.

## 1. Introduction

Obstructive sleep apnea (OSA) is a sleep-related breathing disorder characterized by recurrent episodes of partial or complete upper airway obstruction during sleep, and is highly prevalent in the general population [1]. The prevalence of obstructive sleep apnea is conservatively estimated to be 3% among women and 10% among men 30 to 49 years of age and 9% among women and 17% among men 50 to 70 years of age [2]. The global prevalence of OSA was reported as almost 1 billion people [3]. Several studies involving Japanese patients with OSA suggested that this disorder could be related to daytime sleepiness, depression, anxiety symptoms, and locomotive syndrome [4,5,6].

An efficient treatment for OSA is continuous positive airway pressure (CPAP), demonstrated to improve daytime symptoms and to reduce cardiovascular disease [7]. Other treatments are positional therapy [8] (avoiding a supine sleep position), surgical management [9], and weight management, including lifestyle interventions with dietary modification and physical activity, pharmacotherapy, and bariatric surgery [10].

Oral appliance (OA) therapy is a treatment option for patients with OSA. The treatment mechanism of OA is to advance the mandible in order to reduce the collapsibility of the upper airway during sleep [11,12]. In Japan, OA therapy for OSA became covered by health insurance in the field of dentistry in 2004. In 2014, treatment guidelines regarding oral appliances for obstructive sleep apnea syndrome recommended the use of OAs that advanced the mandible forward and limited mouth opening for patients with OSA. However, CPAP should be used by patients for whom it has been indicated. OAs are desirable for those who cannot use CPAP (GRADE 1B, strong recommendation/quality of evidence, “moderate quality”), which was published by the Japanese Academy of Dental Sleep Medicine [13], and OA therapy has been widely applied. Currently, OA therapy is performed at many dental clinics.

In Japan, OA therapy is covered by health insurance. It was reported that the number of non-obese patients with OSA among Asians, including the Japanese, was greater than that among Caucasians [14]. Thus, in Japan, a special medical insurance system and race-related characteristics are present, but the background of patients undergoing OA therapy and treatment results remain to be clarified. In this study, a multicenter research group is established by the Japanese Academy of Dental Sleep Medicine, and the Japanese cross-sectional multicenter survey (JAMS) of OA therapy is conducted to investigate the current status of OA therapy, including efficacy, side effect, and follow-up in Japan.

## 2. Materials and Methods

### 2.1. Subjects and Survey Period

We investigated 10 medical institutions where the annual number of patients with OSA who initially consulted the Department of Dentistry was ≥50: four university hospitals (Osaka University Dental Hospital, Aichi Medical University, Tokyo Medical and Dental University, Nihon University School of Dentistry at Matsudo), three general hospitals (Ota Memorial Sleep Center, JR Tokyo General Hospital, Komatsu Hospital), and three dental clinics (Katahira Dental Clinic, Furuhata Dental Clinic, Iida Dental Clinic). OA therapy was undertaken by dental clinics that had a board-certified dentist of the Japanese Academy of Dental Sleep Medicine or the Japanese society of Sleep Research. We retrospectively examined the course of treatment until 31 December 2016 in patients who initially consulted the Department of Dentistry at each medical institution between 1 January 2014 and 31 December 2015.

### 2.2. Survey Content

The following items were extracted from the subjects’ medical records: age at the initial consultation, sex, degree of obesity (body mass index (BMI)), examination procedure/respiratory disturbance index at the time of OSA diagnosis, shape of the OA, presence of adjustment after wearing an OA, presence of adverse reactions, presence of OA therapy assessment, examination procedures, and respiratory disturbance index.

Examination procedures at the time of OSA diagnosis and OA efficacy assessment were classified into three types: pulse oximetry, out-of-center sleep testing (OCST), and polysomnography (PSG). We confirmed which test was conducted and extracted the oxygen desaturation index (ODI), respiratory event index (REI), and apnea-hypopnea index (AHI) as indices of sleep-disordered breathing.

The shape of the OA is classified into two types: mandibular advancement devices (MADs) and tongue retaining devices (TRDs). There are two types of MAD: mono and bi blocks. In this survey, the shape of the OA was classified into three types: mono blocks, bi blocks, and TRDs. Adjusting the mandible protrusion range of an OA after confirming the effects of/adverse reactions to wearing an OA was defined as adjustment. Pain after wearing an OA was defined as an adverse reaction. The sites of pain included the gingiva, teeth, jaw, masticatory muscle, and temporal muscle. Patients with the spontaneous remission of pain were included in those with adverse reactions. Those with discomfort were excluded.

Among patients for whom assessment with the same examination procedure was possible before and after OA therapy, the treatment response was evaluated. As an index of the treatment response, the rate of decrease in the AHI (%) was calculated. We regarded patients with a posttreatment AHI of <5/h as complete responders, those with an AHI of ≥5/h and a ≥50% decrease in the AHI as partial responders, and those with an AHI of ≥5/h and a <50% decrease in the AHI as non-responders. The rates of the respective responders were calculated.

### 2.3. Survey Methods

For data input, a common work sheet was delivered to each institution. Regarding ethical considerations, the data were submitted through dummy coding of the patient name and chart ID number such that each patient was unable to be identified. The main author (Kentaro Okuno) (chief of the study project) was responsible for data collection/statistical analysis. The validity of the results of the statistical analysis was approved by all investigators. The protocol of this study was approved by the ethics committee of Osaka University Dental Hospital (H28-E20).

### 2.4. Statistical Analysis

All data were analyzed by SPSS 15.0 statistical software (SPSS Inc., Chicago, IL, USA). Descriptive statistics for clinical characteristics were presented as mean ± SD. Continuous variables were evaluated with an unpaired t-test to compare the adult (under 65 years) and elderly (over 65 years) groups, based on the definition of elderly by WHO. A two-way analysis of variance (ANOVA) assessed the differences in age, BMI, and AHI between the types of institution, and the differences in age, BMI, AHI, and the AHI reduction rate between the severities of AHI. When the analysis of variance showed a *p*-value less than 0.05, comparisons were performed using the Bonferroni correction. A *p*-value of <0.05 was considered to indicate statistical significance.

## 3. Results

The total number of patients who consulted the Department of Dentistry was 3217: 1546 at university hospitals, 560 at general hospitals, and 1111 at dental clinics. Of these, 2947 underwent OA therapy. The remaining 270 patients selected treatment methods other than OA therapy or did not request treatment. For 1600 patients, the efficacy of OA therapy was evaluated, whereas it was not evaluated or treatment/consultation was discontinued for 1347 patients. For 1050 of those evaluated, the same examination procedure was performed before and after treatment.

The basic information of patients who underwent OA therapy is shown in Table 1. The mean age was 52.7 ± 13.8 years. Males accounted for 75.2%. The mean body mass index (BMI) was 23.9 ± 3.5 kg/m^2^. The mean age of those at university hospitals was 55.3 ± 14.0 years, being significantly higher than at general hospitals (52.4 ± 12.9 years) and dental clinics (49.6 ± 13.4 years). Regarding age, the number of patients aged 50 to 59 years was the largest (*n* = 752), followed by that of those aged 40 to 49 years (*n* = 673), and that of those aged 60 to 69 years (*n* = 611). The subjects included one child aged <10 years and two patients aged ≥90 years (Figure 1). Among males, there were 546 aged 40 to 49 years and 550 aged 50 to 59 years. Among females, there was a rapid increase in the number of patients from 40 to 49 years of age, reaching a peak at 50 to 59 years of age (*n* = 202) (Figure 1). The mean AHI at the time of OSA diagnosis was 21.4 ± 15.1/h. Based on the AHI, the severity of OSA was evaluated as simple snoring (snoring, AHI: <5) in 2.3%, mild (5 ≤ AHI < 15) in 38.5%, moderate (15 ≤ AHI < 30) in 39.9%, and severe (30 ≤ AHI) in 19.3%. The mean AHI at the general hospitals was 24.5 ± 10.4/h, being significantly higher than that at the university hospitals (21.9 ± 15.2/h) and dental clinics (19.4 ± 14.4/h). With respect to age, the mean AHI values in the adult (under 65 years) and elderly (over 65 years) groups were 20.7 ± 15.1/h and 23.7 ± 14.7/h, respectively, exhibiting a significant difference. Regarding severity classification, mild-status patients accounted for 41.3% (higher percentage) in the former group, whereas moderate-status patients accounted for 45.1% (higher percentage) in the latter group. Severe-status patients accounted for 17.7% and 24.7%, respectively.

Regarding the type of device, mono blocks were used for 91.8%, bi blocks for 7.9%, and TRD for 0.3%. The rate of patients for whom bi blocks were used at the dental clinics was 17.6%, being higher than that at the university and general hospitals. Adjustment of mandibular advancement after wearing the OA was conducted for 18.6%. It was not performed for most patients. Adverse reactions developed in 14.7% of the subjects. Overall, OA reassessment was conducted for 54.3%. The percentages at university hospitals, general hospitals, and dental clinics were 59.0%, 51.7%, and 49.6%, respectively (Table 2).

Among patients for whom the same examination procedure was adopted before and after treatment, the effects of OA therapy were calculated (Table 3). The mean AHI decreased from 22.4 ± 14.5/h to 9.3 ± 9.2/h, and the mean rate of decrease in the AHI was 52.0% ± 43.7%. Complete response was achieved in 35.6%, and partial response in 31.3%. Non-responders accounted for 33.0%. Concerning the treatment response with respect to severity classification, the rate of complete responders in the severe group was 40.5%, being higher than that of the mild and moderate groups (31.4% and 36.9%, respectively). With respect to age, complete responders accounted for 43.3% in the elderly group. The percentage was higher than that in the adult group (32.9%).

## 4. Discussion

This is the first large-scale survey in Japan regarding the current status of OA therapy, involving 3217 patients with OSA at 10 institutions (four university hospitals, three general hospitals, and three dental clinics).

Regarding the distribution of the number of patients with respect to age, the morbidity rate of sleep-disordered breathing increases with age, reaching a peak at 60 to 69 years of age regardless of sex [15]. In this study, the number of patients peaked at 50 to 59 years of age for both males and females. A peak was noted at an age younger than previously reported, but a similar tendency was observed. For females, the number of patients aged 20 to 29 years or 30 to 39 years was very small, differing from males. There was a rapid increase after 40 years of age. The Wisconsin Sleep Cohort Study reported that the incidence of OSA in postmenopausal women was higher than in premenopausal women [16]. There was also a rapid increase in the number of female patients after 40 years of age in this survey. This may have been related to female-specific hormonal imbalance or menopause, in addition to aging-related muscular weakness or fat deposition.

In patients who underwent OA therapy, the severity of OSA was evaluated as simple snoring in 2.3%, mild in 38.5%, moderate in 39.9%, and severe in 19.3%. OA therapy is indicated for patients with mild to moderate OSA or severe OSA patients for whom the use of a continuous positive airway pressure (CPAP) device is difficult [17]. Of the subjects of this survey, severe OSA was observed in 19.3%, therefore, this survey was performed without focusing on mild-status patients. With respect to institutions, the number of moderate and severe OSA patients was high at the university and general hospitals, whereas that of mild and moderate OSA patients was high at dental clinics. This suggests that more severe-status patients consult university or general hospitals.

Regarding the success rate of OA therapy, complete responders, partial responders, and non-responders accounted for 35.6%, 31.3%, and 33.0%, respectively, in this survey. According to a previous systematic review on OA therapy, the mean rate of patients with an AHI of <5, which was the same criterion adopted in this survey, was 48%. The success rate was higher than that in this survey [12]. In some articles included in this systematic review, the subjects were limited to mild or moderate OSA patients. Furthermore, there have been many prospective studies involving a specific patient group consenting to study participation, therefore, the patient background may have differed from that of the subjects of this survey, which may have led to a difference in the treatment results. The mean rate of patients with a ≥50% decrease in the AHI, corresponding to partial responders in this survey, was 35%, which was similar to that of the previous survey (31.3%).

According to a previous review on adverse reactions, the incidence ranged from 6% to 86% [18]. The types of adverse reactions varied from severe symptoms, such as pain of the jaw and tooth pain, to relatively mild symptoms, such as discomfort on wearing, hypersalivation, and discomfort on occlusion. In this survey, adverse reactions that may have influenced treatment adherence, such as pain of the gingiva, teeth, jaw, masticatory muscle, and temporal muscle, excluding discomfort, were investigated. As a result, the incidence of adverse reactions was 14.7%.

Regarding the shape of the OA, mono blocks were used for 91.8% of the patients, followed by bi blocks (7.9%), and TRDs (0.3%). TRDs were less frequently selected, possibly because there were few reports on their effects. In Japan, OA therapy is covered by dental health insurance. As medical fee points are established, bi blocks of MADs, which require a high technical fee, or TRDs are difficult to select, and this may have led to the use of mono blocks of MADs for most patients.

Mandibular advancement after wearing an OA was adjusted for 18.6% of the subjects. Adjustment was not conducted for most patients. Dort [19] emphasized that, even when adopting a mono block, it is necessary to separate the upper and lower parts and fix them again in order to be effective. It may be important for dentists to recognize the necessity of adjusting mono-block-type OAs, which are routinely used in Japan.

In this survey, the OA reassessment rate was 54.3%, therefore, treatment was unable to be evaluated for approximately 50% of the subjects after OA therapy. As one of the reasons for this low rate, the treatment response cannot be evaluated at dental clinics due to the dental health insurance system. As a result, it is necessary to additionally consult a medical institution after wearing an OA. In some cases, OA adjustment is performed based on the results, requiring a third assessment. The health expenditure is high and it is time consuming, which is an issue. To overcome this issue and reduce patient stress, OA efficacy assessment at dental clinics should be approved. The reassessment rate at university hospitals was 59%, being higher than that at general hospitals and dental clinics (51.7% and 49.6%, respectively). At university hospitals, close relationships with in- and out-of-hospital medical departments may have played a role in the high reassessment rate.

In this survey, non-responders accounted for 33%. Thus, the treatment response may be insufficient in 1 of 3 patients for whom treatment is not evaluated. Such patients may lose the opportunity of receiving OA adjustment or other treatment methods, including CPAP therapy, and severe OSA may remain untreated due to insufficient therapeutic effects. In the future, the OA reassessment rate should be improved.

This survey has the following limitations: it was a retrospective investigational study, and it was impossible to standardize a protocol for OA therapy among the institutions; only items that were common among the institutions were investigated. As a result, we were unable to examine the Epworth Sleepiness Scale (ESS) score, OA wearing rate, or details of adverse reactions. Furthermore, this survey involved medical institutions at which the annual number of OSA patients who initially consulted the Department of Dentistry was ≥50, therefore, institutions in urban areas comprised the greater portion, and the results may not reflect the current status in non-urban areas.

## 5. Conclusions

This survey revealed the current status of OA therapy in the management of obstructive sleep apnea, of which the rate of AHI < 5 with OA was 35.6%, adverse reactions developed in 14.7%, and OA reassessment was conducted for 54.3% in Japan. In the future, a prospective, multicenter study with a standardized treatment protocol should be conducted.

## Figures and Tables

**Figure 1 ijerph-16-03288-f001:**
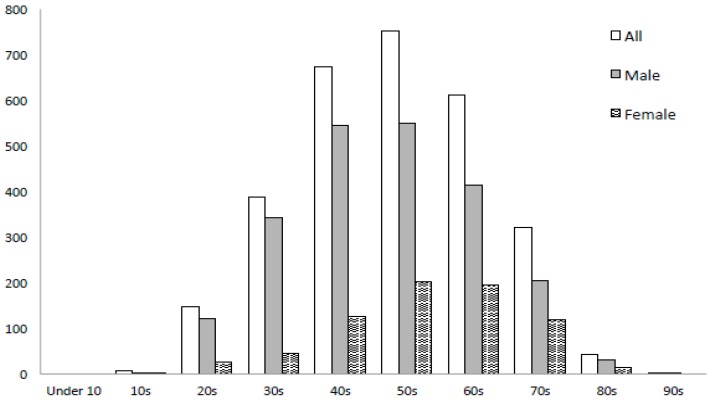
Age distribution of the patients.

**Table 1 ijerph-16-03288-t001:** Patient characteristics.

Variable	All	University Hospital	General Hospital	Private Clinic	*p*	Age < 65	Age ≥ 65	*p*
N (*n*)	2947	1344	536	1067		2286	661	
Age (Y)-mean ± sd	52.7 ± 13.8	55.3 ± 14.0	52.4 ± 12.9 *	49.6 ± 13.4 *^,^ **	0.000 ^†^			
Male-*n* (%)	2216 (75.2%)	948 (70.5%)	422 (78.7%)	846 (79.3%)		1789 (78.3%)	427 (64.6%)	
Female-*n* (%)	731 (24.8%)	396 (29.5%)	114 (21.3%)	221 (20.7%)		497 (21.7%)	234 (35.4%)	
BMI (kg/m2)-mean ± sd	24 ± 5.5	24.1 ± 3.7	24.5 ± 10.4	23.6 ± 3.4	0.674 ^†^	24.1 ± 3.7	23.8 ± 9.4	0.180 ^††^
AHI (/h)-mean ± sd	21.4 ± 15.1	21.9 ± 15.2	24.0 ± 1 5.7 *	19.4 ± 14.4 *^,^ **	0.000 ^†^	20.7 ± 15.1	23.7 ± 14.7	0.000 ^††^
AHI severity								
Snoring-*n* (%)	69 (2.3%)	25 (1.9%)	5 (0.9%)	39 (3.7%)		61 (2.7%)	8 (1.2%)	
Mild-*n* (%)	1133 (38.5%)	510 (38.1%)	172 (32.1%)	451 (42.3%)		942 (41.3%)	191 (29.0%)	
Moderate-*n* (%)	1172 (39.9%)	522 (39.0%)	224 (41.8%)	426 (39.9%)		875 (38.4%)	297 (45.1%)	
Severe-*n* (%)	566 (19.3%)	283 (21.1%)	135 (25.2%)	148 (13.9%)		403 (17.7%)	163 (24.7%)	

BMI, body mass index; AHI, apnea and hypopnea index; Snoring, AHI < 5; Mild, 5 ≤ AHI < 15; Moderate, 15 ≤ AHI < 30; Severe, AHI ≥ 30; †, Two-way analysis of variance (ANOVA); ††, T-test; *: significant association by the Bonferroni correction (*p* < 0.05) versus University Hospital; **: significant association by the Bonferroni correction (*p* < 0.05) versus General Hospital.

**Table 2 ijerph-16-03288-t002:** Oral appliance type and evaluation methods.

Variable	All		University Hospital		General Hospital		Private Clinic	
N (*n*)	2947		1344		536		1067	
Mono block-*n* (%)	2705	(91.8%)	1294	(96.3%)	532	(99.3%)	879	(82.4%)
Bi block-*n* (%)	234	(7.9%)	46	(3.4%)	4	(0.7%)	188	(17.6%)
TRD-*n* (%)	8	(0.3%)	4	(0.3%)	0	(0%)	0	(0%)
Adjustment	548	(18.6%)	298	(22.2%)	17	(3.2%)	233	(21.8%)
Adverse reactions	434	(14.7%)	222	(16.5%)	61	(11.4%)	151	(14.2%)
OA follow-up sleep study	1599	(54.3%)	793	(59.0%)	277	(51.7%)	529	(49.6%)
Method of diagnosis for OSA								
PSG-*n* (%)	1792	(60.9%)	663	(49.5%)	399	(74.4%)	730	(68.4%)
OCST-*n* (%)	1122	(38.1%)	674	(50.3%)	130	(24.3%)	318	(29.8%)
Pulse oximetry-*n* (%)	29	(1.0%)	3	(0.2%)	7	(1.3%)	19	(1.8%)
Method of evaluation for OA								
PSG-*n* (%)	491	(30.7%)	189	(23.8%)	190	(68.6%)	112	(21.2%)
OCST-*n* (%)	1065	(66.6%)	568	(71.6%)	86	(31.0%)	411	(77.7%)
Pulse oximetry-*n* (%)	43	(2.7%)	36	(4.5%)	1	(0.4%)	6	(1.1%)

OA, oral appliance; OSA, obstructive sleep apnea; PSG, polysomnography; OCST, out-of-center sleep testing.

**Table 3 ijerph-16-03288-t003:** OA efficacy.

Variable	All	Snoring	Mild	Moderate	Severe	*p*	Age < 65	Age ≥ 65	*p*
N (*n*)	1050	6	354	480	210		773	277	
Age (Y)-mean ± sd	54.9 ± 13.2	48.5 ± 11.2	53.0 ± 12.7	55.9 ± 13.3 *	56.1 ± 13.6 *	0.004 ^†^			
Male-*n* (%)	784 (74.3%)	2 (33.3%)	245 (69.2%)	362 (75.4%)	175 (83.3%)		607 (78.5%)	100 (36.1%)	
Female-*n* (%)	266 (25.3%)	4 (66.7%)	109 (30.8%)	118 (24.6%)	35 (16.7%)		166 (21.5%)	177 (63.9%)	
BMI (kg/m2)-mean ± sd	23.9 ± 3.5	23.5 ± 2.3	23.2 ± 3.1	23.8 ± 3.5	25.4 ± 3.8 *^,^ **	0.000 ^†^	24.1 ± 3.7	23.4 ± 3.0	0.004 ^††^
Before AHI (/h)-mean ± sd	22.4 ± 14.5						22.0 ± 14.9	18.8 ± 12.2	0.144 ^††^
After AHI (/h)-mean ± sd	9.3 ± 9.2	8.5 ± 4.8	10.3 ± 9.3	9.0 ± 9.4	8.3 ± 8.4	0.058 ^†^	9.9 ± 9.6	7.7 ± 7.7	0.176 ^††^
AHI reduction rate (%)-mean ± sd	52.0 ± 43.7	51.3 ± 55.2	49.2 ± 37.4	56.0 ± 38.3	47.9 ± 61.0	0.062 ^†^	52.5 ± 38.5	50.7 ± 55.8	0.112 ^††^
OA treatment response									
Complete responder-*n* (%)	374 (35.6%)		111 (31.4%)	177 (36.9%)	85 (40.5%)		254 (32.9%)	120 (43.3%)	
Partial responder-*n* (%)	329 (31.3%)		102 (28.8%)	165 (34.4%)	59 (28.1%)		254 (32.9%)	75 (27.1%)	
Non-responder-*n* (%)	347 (33.0%)		141 (39.8%)	138 (28.8%)	66 (31.4%)		265 (34.3%)	82 (29.6%)	

BMI, body mass index; AHI, apnea and hypopnea index; OA, oral appliance; Snoring, AHI < 5; Mild, 5 ≤ AHI < 15; Moderate, 15 ≤ AHI < 30; Severe, AHI ≥ 30; Complete responder, AHI < 5; Partial responder, AHI reduction rate of ≥50% and AHI of ≥5; Non-responder, AHI reduction rate of ≥50%; †, Two-way analysis of variance (ANOVA); ††, T-test; *: significant association by the Bonferroni correction (*p* < 0.05) versus Mild; **: significant association by the Bonferroni correction (*p* < 0.05) versus Moderate.

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
