# Peer review of "Japanese Cross-Sectional Multicenter Survey (JAMS) of Oral Appliance Therapy in the Management of Obstructive Sleep Apnea"

_ijerph, 2019, doi:10.3390/ijerph16183288_

Round 1
Reviewer 1 Report
TITLE: The abbreviation JAMS should be moved after "Japanese Multicenter Survey".
The title itself does not explain what is the purpose of the study.
ABSTRACT:
line 25-26: Please specify the aim of the study. The sentence looks vague.
I suggest to report the article following the STROBE checklist.
INTRODUCTION:
Please, clarify the indications/guidelines for OA according to Japanese Academy of Dental Sleep Medicine.
Please, clarify in bullet points what are you looking at.
MATERIALS AND METHODS:
Please, add informations regarding the clinical management of OA: who are the clinicians (general dentists, orthodontists), which adjustments were required.
Survey content: I think you should add some data (education, smoking status, alcoholism, wearing time of OA ) to subjects' medical records; please, refer to the study below:
Romandini M, Gioco G, Perfetti G, Deli G, Staderini E, Lafori A. The association between periodontitis and sleep duration. J Clin Periodontol. 2017 May; 44(5):490-501. doi: 10.1111/jcpe.12713. Epub 2017 Apr 19.
line 73: there is a space between OA and the coma
DISCUSSION:
Please make comparisons with other studies and different countries, emphasizing the role of OA in the management of AHI.
CONCLUSION:
The paragraph must be improved. You should answer to a research question.
REFERENCES:
Please checkthe typo in reference #3: "Obstruc-tive"
Reviewer 2 Report
The authors of this Japanese survey of the use of oral appliance in obstructive sleep apnea showed interesting data, but the draft needs to improve some important aspects for the publication.
Title: In the title appears the abbreviation JAMS that I imagine will be the acronym of the study. This must be explained somewhere since the first thing I thought was that it was the type or trademark of a device. Title: I suggest that the authors delete the number of patients. Please include this information in results Abstract: Please define the abbreviation TRD Introduction: Authors should update OSA prevalence data. The reference they have for data in Japan was more than 15 years ago. Please update. The authors can review the following article: Adam V Benjafield, Najib T Ayas, Peter R Eastwood, et al. Estimation of the global prevalence and burden of obstructive sleep apnoea: a literature-based analysis. Lancet Respir Med 2019; 7: 687–98 The introduction is too short, and it is not possible to understand what the therapeutic alternatives are for SAHOS and in which case the oral devices would be justified. I suggest adding a paragraph as number two (between epidemiology and oral devices) explaining the gold standard (positive pressure), and the therapeutic alternatives: Surgery, exercise, diet, oral devices. The authors can review the following updated references (2019): Veasey SC, Rosen IM. Obstructive Sleep Apnea in Adults. N Engl J Med. 2019 Apr 11;380(15):1442-1449. doi: 10.1056/NEJMcp1816152. Mandavia R, Mehta N, Veer V. Laryngoscope. Guidelines on the surgical management of sleep disorders: A systematic review. 2019 May 1. doi: 10.1002/lary.28028. Torres-Castro R, Vilaró J, Martí JD, Garmendia O, Gimeno-Santos E, Romano-Andrioni B, Embid C, Montserrat JM. Effects of a Combined Community Exercise Program in Obstructive Sleep Apnea Syndrome: A Randomized Clinical Trial. J Clin Med. 2019 Mar 14;8(3). pii: E361. doi: 10.3390/jcm8030361. Tham KW, Lee PC, Lim CH. Weight Management in Obstructive Sleep Apnea: Medical and Surgical Options. Sleep Med Clin. 2019 Mar;14(1):143-153. doi: 10.1016/j.jsmc.2018.10.002. Methods: Why did you decide to make the cut-off point in 65 years? Please explain. Methods: Please, describe the design of the study I suggest adding the Survey as complementary information (ideally in English)Author Response
Please see the attachment

Round 2
Reviewer 2 Report
The authors have made the suggested changes satisfactorily. I consider that it is in a position to be published.